# Experimental study on the coupling control of rock blasting vibration and rock fragmentation size

**Li He[1,2], Wuyi Zhang[2]\*, Yongming Zhao[2], Sheng Peng[2], Yingkang Yao[1], Shasha Chen[2], Maolin Wang[2]**

**1** Hubei Key Laboratory of Blasting Engineering, Jianghan University, Hubei, Wuhan, China, **2** College of Physics and Mechanics, Wuhan University of Science and Technology, Hubei, Wuhan, China

\* 18972228852@163.com

## Abstract

To address the issue that open-pit blasting delay designs struggle to concurrently mitigate vibration and optimize fragmentation, this article proposes an interval-based delay optimization framework integrating theory and experiments. Unlike traditional fixed-value delay designs, this framework responds to the variability of blasting parameters and rock mass properties, determining a reasonable delay time interval rather than a single fixed value to realize synergistic control of vibration mitigation and fragmentation optimization. It adopts Hanukayev's theory for delay time calculation, the Swebrec function for fragmentation evaluation, and the "vibration reduction rate" for vibration assessment. Experiments were conducted on C50 concrete specimens using a 3-hole model. The results indicate that inter-hole delays of 3-4 ms and inter-row delays of 1-3 ms enable effective coupling control. Among all schemes, the optimal 3 ms/3 ms (inter-hole/inter-row) scheme achieves remarkable effects: it reduces the peak particle velocity (PPV) in the free surface direction and reverse direction by 70% and 54% respectively, while decreasing the average fragment size and large fragment size by 10% and 9.3% respectively. This research provides crucial theoretical support for safe and efficient mine blasting operations.

## 1. Introduction

Open-pit blasting, as a core technology in mining engineering and infrastructure excavation, its operational effect is directly linked to engineering safety and construction efficiency-excessive blasting vibration tends to damage surrounding structures, geological hazards and other safety risks, while uneven rock fragmentation size will increase excavation and transportation energy consumption and reduce the efficiency of subsequent processes. There is often a coupling conflict regarding the challenge of balancing vibration control and fragmentation optimization [1–5]. How to achieve the homogenization and rationalization of rock fragmentation size while suppressing

**Data availability statement:** All relevant data are within the paper and its Supporting Information files.

**Funding:** This work was funded by the general project of Hubei Provincial Key Laboratory of Blasting Engineering(BL2021-11),National Natural Science Foundation of China(52274136,51904210), and Natural Science Foundation of Hubei Province(2024 AFB766). We thank the Wuhan University of Science and Technology for providing experimental sites and equipment. The funders had no role in study design, data collection and analysis, decision to publish, or preparation of the manuscript.

**Competing interests:** The authors have declared that no competing interests exist.

excessive blasting vibration is a key problem that has long been urgently solved in the field of open-pit blasting engineering. Based on this, this study focuses on the coupling control mechanism of rock blasting vibration and rock fragmentation size, and constructs a delay time optimization framework through laboratory experiments, aiming to provide theoretical support and technical reference for balancing blasting safety and operational efficiency.

To realize reasonable control of blasting vibration, researchers have used theoretical inference, numerical simulation, model construction, field tests, and other research methods to analyze the impact of different vibration reduction measures on blasting vibration [6–7]. Zhu et al. further verified the dominant effect of geotechnical material spatial variability on slope failure risk via probabilistic stability analysis of two-layer undrained slopes, which highlights the engineering necessity of precise blasting vibration control in open-pit projects [8]. Millisecond delay blasting is the core of blasting and vibration reduction technology, and the optimization of the delay time is the most effective way to give full play to the advantages of digital electronic detonators. For example, Liu et al. used the hole-by-hole detonation method with multiple rows of holes delayed in groups to effectively utilize the blasting energy, improve the blasting effect, and simultaneously successfully control the blasting vibration, flying rocks, and other hazards to ensure the safety of the surrounding area of the explosion [9]. Huang et al. found that blast delays shorter than 6 ms yield poor vibration reduction effects, particularly in the near-field region. As the delay time increases, the overall amplitude reduction ratio (ARR) gradually rises. Therefore, extending the delay time to 8–10 ms in actual blasting operations helps further reduce the misfire rate [10]. However, this study only focused on the vibration reduction effect of short-delay blasting and did not explore the corresponding impact on crushing fragmentation uniformity, resulting in a one-sided focus on single safety indicators and failure to meet the dual requirements of engineering safety and operational efficiency. Liu et al found that the effect of millisecond delay delay time on the near zone blasting vibration is much larger than that of the far zone, and the near zone blasting vibration exhibits an initial increase, followed by a decrease and then a subsequent increase while the fluctuation amplitude of the far zone blasting vibration is significantly reduced [11]. Iwano et al. used the superposition method to derive the optimum delay time and applied it to engineering practice. They found that the blasting vibration was reduced when the delay time interval was equal to half of the peak frequency, and that the method has been widely popularized in other blasting projects [12]. Gou et al. used an instantaneous energy method based on empirical mode decomposition (EMD) to identify the actual delay time from the near-field acceleration, and then analyzed the vibrational attenuation using Theil-Sen regression; they found that the peak particle velocity (PPV) and the average frequency (AF) were more sensitive to a shorter delay time, and both were affected by it progressively less with an increase in the delay time.This study provided a theoretical basis for short delay time optimization of blasting vibration [13]. Wang et al. derived a blasting vibration duration prediction formula based on the close connection between the delay time, blasting vibration duration, and favorable frequency, which provides a

new method for calculating the time-domain and frequency-domain expressions of blasting vibration and the power spectral density of the blasting vibration for single- or multi-stage blasting to avoid resonance of the structure [14]. Zhang et al. found that the proportion of low-frequency vibration increased and the dominant frequency shifted to lower frequencies at very short delay times (≤5 ms); At shorter delay times used (6–12 ms), the low frequencies shifted to higher frequencies, which reduced the energy ratio of the low-frequency bands; meanwhile, the use of short delays in the near zone (6–8 ms), and long delays in the mid- and far-range zones (10–20 ms) could better reduce the blast vibration [15]. Zhang et al. and Lin et al. showed through blasting experiments that millisecond delay blasting based on electronic detonators can reduce blasting vibration, but when the delay time reaches a certain value, the blasting vibration will no longer continue to decrease and instead tends to approach the peak vibration intensity of single-hole blasting [16–17]. In summary, scholars both domestically and internationally have demonstrated that short-delay detonation based on digital electronic detonators significantly reduces blasting vibrations and controls the dominant frequency in the near zone of blasting operations.

The rock fragmentation generated by blasting has always been the focus of attention in the engineering field, and the size of rock fragmentation is directly related to the efficiency of subsequent excavation, transportation, and other operations. Domestic and foreign scholars have improved the fragmentation size control theory by means of finite element software, blasting simulation software, simulation experiments, etc., and have achieved many results in controlling the rock fragmentation size [18–21]. For example, Li et al. investigated the fracture evolution and mechanical deterioration of granite under cyclic thermal and liquid nitrogen cryogenic impact, revealing the coupled damage mechanisms of rock mass fracture development and mechanical property degradation, which provides important experimental insights for understanding the rock dynamic fragmentation mechanism in blasting engineering [22]. Saadatmand Hashemi et al. used the LS-DYNA numerical program to study the effect of detonation time on the damage of blasting rock fragmentation and found that the appropriate delay time helps the growth of fissures, thereby obtaining a better rock fragmentation effect [21,23]. Ye et al. used ANSYS/LS-DYNA software to establish the model, carry out the optimization of the calculation model and combined with the demonstration of engineering tests, the results of the study show that there is an exponential function relationship between the size of the blasting chunks and the damage value of the numerical model, and within a certain range, the length of the delay time between the holes and the rate of blasting fragments of the large chunks of the rate of proportional to the rate of pulverization of the powder ore rate inversely proportional to the rate [24]. Tang et al. propose that when optimizing delay time to achieve optimal rock fragmentation, the delayed superposition effect of shock waves and their interaction with crack propagation should be considered [25]. Choudhary et al. found that group-controlled blasting with inter-row delay times of 8−14 ms/m and 5 ms/m for the first, second, and last two rows, respectively, could optimize rock fragmentation [26]. Ma et al. used a three-parameter GEV function to describe the FSDs of blast-induced rock rupture and found that in the case of two-hole blasting, the kinetic and internal energies increased simultaneously with the extension of the delay time, and the kinetic energy continued to be prolonged for a longer period of time, which led to a gradual increase in the degree of rock damage [27]. Zhang conducted systematic studies on the influence of inter-hole and inter-row delay times on blasting fragmentation in open-pit deep hole blasting, optimizing delay parameters to reduce large block rate and maximum fragment size, but failed to incorporate blasting vibration constraints into the optimization framework, making it difficult to directly apply the results to engineering scenarios requiring both safety and efficiency [28,29].

In summary, scholars worldwide have conducted numerous valuable studies on the effect of delay time on blasting vibration and rock fragmentation, which provides important theoretical guidance and practical reference for open terrace blasting operations. By setting a reasonable delay time, the blasting vibration can be better controlled or the rock crushing effect can be optimized. However, most of the current delay-time influence analysis studies focus on a single factor, and there are few comprehensive studies on the synergistic effect of blasting vibration and rock fragmentation.

This study takes this as an entry point to fill this academic gap, and proposes an optimal delay time selection method to achieve the coupled control of reducing blasting hazards and optimizing blasting effect, which provides a solid support

for the optimization of the development of blasting engineering. This study's interval-based delay optimization framework follows four key steps: calculating initial inter-hole and inter-row delay times via Hanukayev's theory; fixing the initial inter-row delay time to screen the optimal inter-hole delay time through comprehensive vibration and fragmentation evaluation; fixing the optimal inter-hole delay time to screen the optimal delay combination via the same method; and adjusting the laboratory delay interval for field verification.It has an important theoretical and practical value for improving the quality of the project, reducing project risk, and helping to promote rock blasting and excavation projects to achieve safe, efficient, and sustainable development. This methodology resolves the core contradiction between vibration safety and fragmentation economics.

## 2. Test progress

This study used 20×20×20 cm homogeneous C50 concrete blocks for testing, with three key differences from actual open-pit blasting: first, the experimental specimens are homogeneous concrete materials without natural joints; second, the size of the experimental specimens is limited by the laboratory scale; third, the laboratory provides a standard and controllable blasting environment and signal acquisition environment. The comparison of mechanical parameters between C50 concrete and jointed gneiss is presented in Table 1.

This study is applicable to small-scale blasting of homogeneous brittle materials (e.g., rock masses with Kv ≥ 0.8 like roadway excavation), providing delay time trend references. For large open-pit deep-hole blasting, on-site parameter calibration is required.

### 2.1 Inter-hole delay time preference test

The purpose of this experiment was to select a time delay scheme that balances the control of blasting vibration and the optimization of fragmentation effect, and to carry out an experimental study on a concrete specimen with a 3-hole blasting model using the controlled variable method. The experimental site and vibration measurement points are shown in Fig 1. Zhang et al. simulated group hole blasting using a three-hole blasting experiment and speculated a reasonable delay time, which proved the rationality of the three-hole blasting experiment in engineering practice [27]. The geometry of the standard concrete specimen was 200 mm×200 mm×200 mm. The concrete specimen mix ratio was cement, sand, stone, and water 1:1.92:3.41:0.54. After pouring, the concrete specimen was cured in (20±2℃) environment for 28 days, the strength grade of C50.The the diameter of the holes was set to 10 mm, the hole spacing was 100 mm, the row spacing was 70 mm, and the depth of the holes was 150 mm. The specimen was then buried in a 200 mm deep pit with its upper surface flush with the ground. Two holes were arranged on one side of the specimen, and this side was set as the free face, and the remaining three sides were compacted with soil. The experimental site is shown in Fig 1(a).

Vibration sensors were arranged in two directions: the free face direction and the free face reverse direction. The sensors used were Blast-NET type blasting seismometers produced by Taitec, and their layout is shown in Fig 1(b).

Table 1. Comparison of mechanical parameters between C50 concrete and jointed gneiss.

| Mechanical Parameters | C50 Concrete | Jointed Gneiss |
|---|---|---|
| Density (ρ), kg/m³ | 2450 | 2600 |
| Elastic Modulus (E), GPa | 33.5 | 35.2 |
| Poisson's Ratio (μ) | 0.20 | 0.25 |
| Uniaxial Compressive Strength ($f_n$), MPa | 50 | 60.5 |
| Tensile Strength (T), MPa | 4 | 7.4 |
| Longitudinal Wave Velocity (C), m/s | 3250 | 3420 |

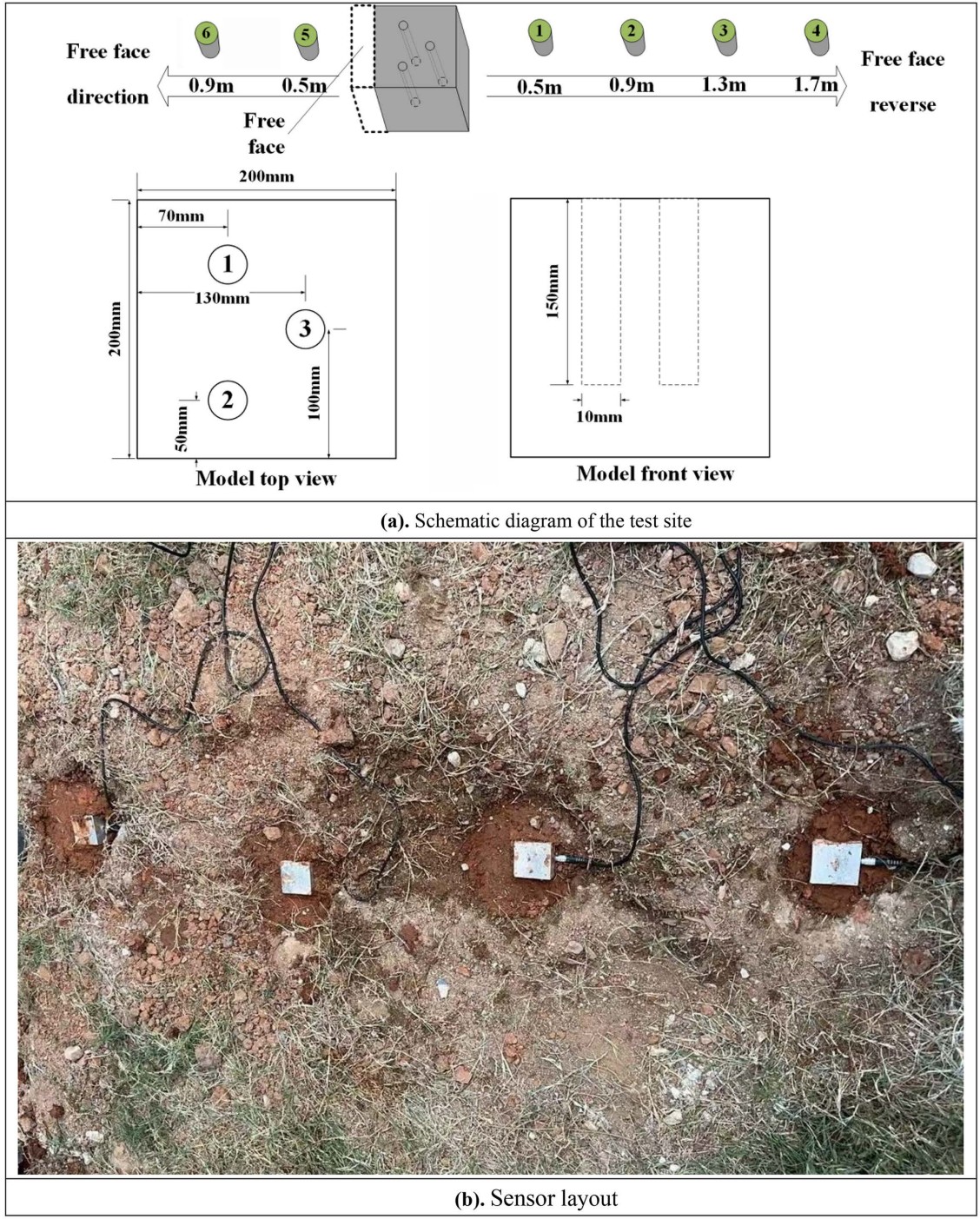

**Fig 1. Schematic diagram of the test site and vibration measurement points.** (a) Schematic diagram of the test site. (b). Sensor layout.

To investigate the vibration effect of delayed blasting, the spacing between adjacent measurement points was set to 0.4 m. Specifically, measurement points 1, 2, 3, and 4 were arranged in the free face reverse direction, and measurement points 5 and 6 were arranged in the free face direction. The distance between measurement points 1, 5 and the test block

was 0.4 m. Concrete test block buried in the soil, in addition to the free face direction of the other three sides by the soil compaction. Test blasting on the three holes in the order of A, B, C shown in Fig 1(a). Because of the small size of the specimen, this experiment only used a digital electronic detonator (equivalent to TNT1.07g) for blasting, that is, after emptying the holes directly into the detonator and plugging tightly with gun clay.

To select the best combination of delay times, this study used the control variable method [26]. By designing multiple groups of blasting experimental groups with different delay times, a comprehensive comparative analysis of the experimental groups on the differences in vibration reduction and fragmentation optimization effects among the groups, screening to determine the optimal combination of delay time between holes. The optimized experimental schemes for inter-hole delay time are listed in Table 2.

### 2.2 Inter-row delay time preference test

To investigate the effect of delay time on blasting vibration and rock fragmentation, based on keeping the delay time between holes unchanged at 3 ms, a new delay scheme is formed with different delay times between rows, and a comprehensive comparison of the experimental results is performed to select the optimal delay scheme that can reduce the vibration of blasting and optimize the effect of rock fragmentation. The experiments set three different rows of delay time and the optimal combination of inter-hole delay time of 3 ms to form a new delay program. Because hole delay of 3 ms, 3 ms between rows of experiments have been completed, the rows of experiments to optimize the delay scheme are listed in Table 3.

## 3. Theoretical derivation

### 3.1 Determination of delay of time

Hanukayev et al. [26] believed that a reasonable delay time should be the time when the blast-generated gas rushes out of the fissure, connects and expands the crushed and ruptured zones, and subsequently forms a blast funnel by first blasting the borehole, and gives a formula for the calculation of the delay time:

$$T = t_1 + t_2 + t_3 = \frac{2W}{C} + \frac{r_c}{V_t} + \frac{S}{V_a}$$

(1)

where $t_1$ represents the time for the stress wave to penetrate to the free face and return, s; $t_2$ represents the formation time of the fissure, s; $t_3$ represents the time for the fragmentation to detach from the parent rock under the action of the

**Table 2. Optimized experimental delay scheme for inter-hole delay time.**

| Groups | 1 | 2 | 3 | 4 | 5 |
|---|---|---|---|---|---|
| Inter-hole delay time $t_k$ | 0 ms | 1 ms | 2 ms | 3 ms | 4 ms |
| Inter-row delay time $t_p$ | 0 ms | 3 ms | 3 ms | 3 ms | 3 ms |

**Note:** Conduct 5 sets of blasting experiments for the above delay program.

**Table 3. Optimized experimental delay scheme for inter-row delay time.**

| Groups | 4 | 6 | 7 | 8 |
|---|---|---|---|---|
| Inter-hole delay time $t_k$ | 3 ms | 3 ms | 3 ms | 3 ms |
| Inter-row delay time $t_p$ | 3 ms | 1 ms | 2 ms | 4 ms |

**Note:** Conduct 3 sets of blasting experiments for the above delay program.

explosive gas, s; $W$ represents the minimum resistance line, m; $C$ represents the longitudinal wave velocity of the rock, m/s; $r_c$ represents the radius of the damage of the stress wave, $m$; $V_t = 0.05C$ represents the fissure expansion speed under the action of the stress wave, m/s; $S$ represents the expansion width of the fissure under the action of explosive gas, $m$; $V_a$ represents the average movement speed of the rock mass, m/s.

The relationship between the charge $\omega$ and radius $r_c$ of the damage zone is

$$r_c = \left( \frac{\omega \rho c^2 Q_v \eta r_e}{S \sigma_c^2 l} \right)^{\frac{1}{2}} \sqrt{\omega}$$

(2)

where $V_a = u_x, u_x$ represents:

$$u_x = \sqrt{\frac{2\pi \rho_e Q_V \eta r_e}{S\rho}}$$

(3)

where $S$ represents the area borne by each shell hole, m²; $\rho_e$ represents the density of the charge, kg/m³; $Q_v$ represents the heat of detonation of the explosive, J; $\rho$ represents the density of the rock (kg/m³; $u$ represents the speed of rock movement, m/s; $r$ represents the radius of the rock rupture zone, m; $l$ represents the length of the charge, m; $\eta$ represents the efficiency of the conversion of the explosive energy to the kinetic energy of the rock.

Combined with the material mechanical parameters and explosive performance parameters, considering that the experimental use of the digital electronic detonator delay time can be accurate to 1 ms, calculated that the experimental inter-row delay time should be set to 3 ms.

## 3.2 Evaluation of blast rock fragmentation size

The main goal of rock fragmentation optimization is to minimize the energy consumption downstream of the mining process when feeding rock piles into the crusher or for grinding. Therefore, the size distribution of the fragments needs to be studied and optimized, and the quality of the fragments needs to be investigated.

In this study, the fitted Ouchterlony's Swebrec function distribution function was used to evaluate the blast fragmentation size [21–22].

$$P(x) = \frac{1}{1 + [ln\left(x_{max}/x\right)/ln(x_{max}/x_{50})]^b}$$

(4)

where $P(x)$ is the screening rate of the crushed material with a screening size of $X$, $X_{50}$ and $X_{max}$ are the 50% and maximum screening rate of the screened size, respectively, and $b$ is the fluctuation exponent. $b$ can be obtained using the least-squares method.

## 3.3 Evaluation of blasting vibration

Millisecond delay blasting offers significant advantages in reducing the hazards of blasting vibrations, improving rock fragmentation, and enhancing the blasting efficiency. Setting an appropriate delay time can reduce the peak vibration velocity, control the primary frequency of blasting vibrations, and allocate blasting energy effectively, thereby avoiding structural damage while optimizing the size of the blasted rock fragments. However, blasting is a transient dynamic process, and the mechanisms by which it fractures rock bodies are highly complex, making it more challenging to determine the optimal delay time for millisecond delay blasting. In this study, the vibration parameters of simultaneous blasting (with inter-hole delay time $t_k$ and inter-row delay time $t_p = 0$ms) serve as a benchmark. The vibration optimization degree was evaluated

by comparing the vibration parameters of millisecond delay blasting. Therefore, the concept of "vibration reduction rate" is introduced [4]:

$$\delta = \frac{v_0 - v}{v_0} \times 100\%$$

(5)

where $v_0$ is the PPV for simultaneous blasting and $v$ is the PPV for different delay schemes.

## 4. Interpretation

### 4.1 Inter-hole delay time data analysis

Statistics of the vibration data from the inter-hole delay optimization experiment and plotting of the PPV and its rate of decrease are shown in Fig 2.

The experimental results showed that the PPV for each delay time group was significantly attenuated at different measurement points compared with simultaneous blasting. PPV decreased with increasing center-of-blast distance. The magnitude and rate of the PPV decay decreased significantly with a further increase in the center distance. In addition, the vibration reduction rate of PPV fluctuated and decreased with the increase of blast center distance with an increase in the center-of-blast distance, which indicates that the effect of vibration damping is most obvious in the near zone, whereas the effect of vibration damping in the far zone is gradually weakened.

In the free-face reverse, by comparing the vibration data of different groups, it can be seen that with an increase in the delay time, the PPV is generally reduced, and the attenuation amplitude tends to increase. Groups 4 and 5 of the PPV difference are very small, indicating that there is a minimization of PPV delay time, which coincides with the theoretical calculation of the optimal delay time of 3 ms rock fragmentation size, verifying the theory of the reality of the basis.

In the free face direction, when the delay time was 3 ms, group 4 had the lowest vibration velocity and the best vibration reduction effect, whereas in group 5, owing to the superposition of waves, the vibration velocity was almost equal to that of simultaneous blasting. The free-face direction of the amplitude of the vibration velocity reduction rate increased with an increase in the burst center distance, whereas group 4 showed a decreasing trend.

In summary, group 4 has the best damping effect, and in view of the practical situation in the field and the need for accurate fragmentation data, the manual sieving method was used to count the blasted fragmentation.

In the field of blasting engineering, $X_{50}$ (median grain size), $X_{80}$ (large grain size) and Xmax (maximum grain size) are the key fragmentation size evaluation indexes. $X_{50}$ is the grain size limit of 50% of the mass in the grain size distribution of the rock mass after blasting, $X_{80}$ indicates the grain size limit of 80% of the mass, which is often used as a reference for the large fragmentation rate, and $X_{max}$ delineates the upper limit of the grain size distribution. The Fragmentation statistics process is shown in Fig 3. The experimental fragmentation size data for inter-hole delay blasting are shown in Table 4.

Plot the Ouchterlony's Swebrec function fitting distribution diagram based on the experimental fragment size data following these steps:1. Collect and sieve fragment samples to obtain cumulative mass percentage P(x); 2. The parameter b is calculated by fitting the cumulative mass percentage data P(x) and the corresponding fragment size x using the least-squares method; 3. Verify the fitting validity with the coefficient of determination $R^2 \geq 0.95$.Ouchterlony's Swebrec function [30] fitted distribution function is shown in Fig 4.

According to Saadatm and Hashemics study, an appropriate delay can optimize the rock crushing effect, but the delay time is too long to reduce the rock crushing efficiency, and theoretically, there should be a delay period that can produce the best rock crushing effect. After analysis, in this experiment, the maximum fragmentation size ($X_{max}$) exhibits an initial increase, followed by a decrease and then a subsequent increase; X80 and X50 exhibit an initial increase followed by a decrease trend. Notably, group 3 ($t_k = 2$ ms) has the highest $X_{max}$ (19.8 cm), mainly due to insufficient stress wave superposition between adjacent holes and inadequate crack expansion caused by short delay time, leading to incomplete

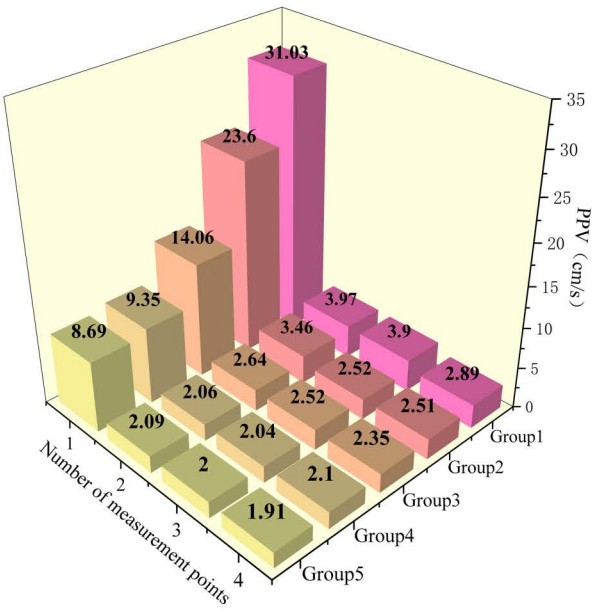

**(a).** Free face reverse PPV

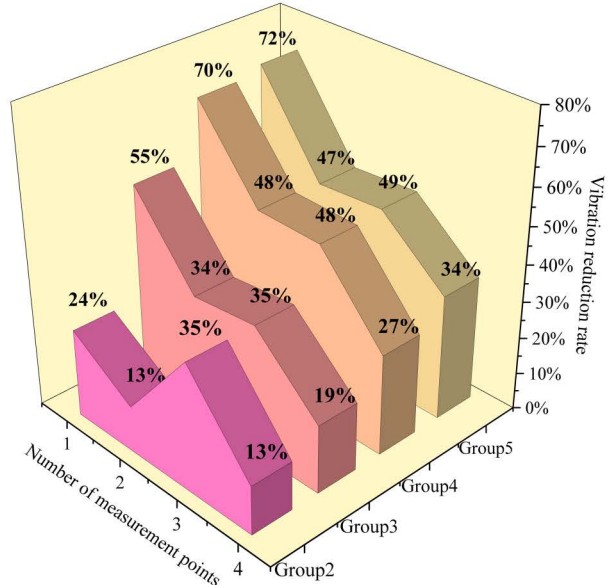

**(b).** Decrease rate of free face reverse PPV

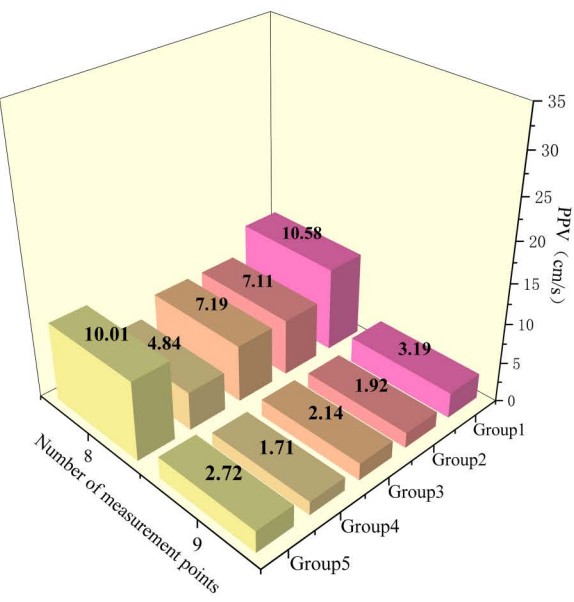

**(b).** Free face direction PPV

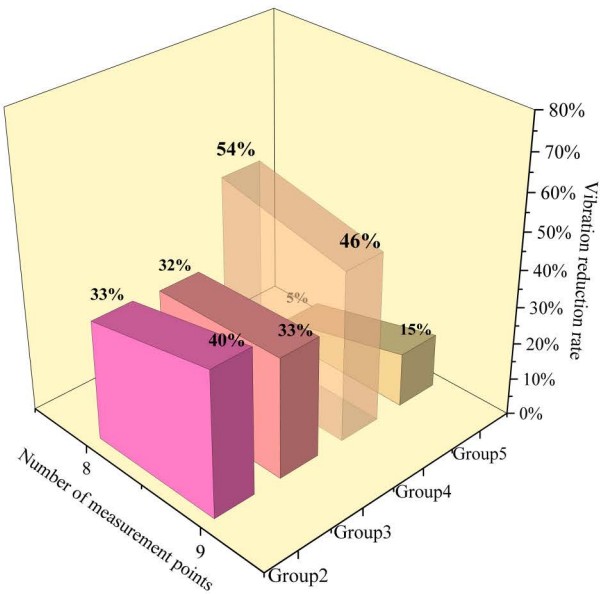

**(d).** Decrease rate of free face direction PPV

**Fig 2. Peak particle velocity (PPV) and vibration reduction rate of different inter-hole delay schemes.** (a) Free face reverse PPV. (b) Decrease rate of free face reverse PPV. (c) Free face direction PPV. (d) Decrease rate of free face direction PPV.

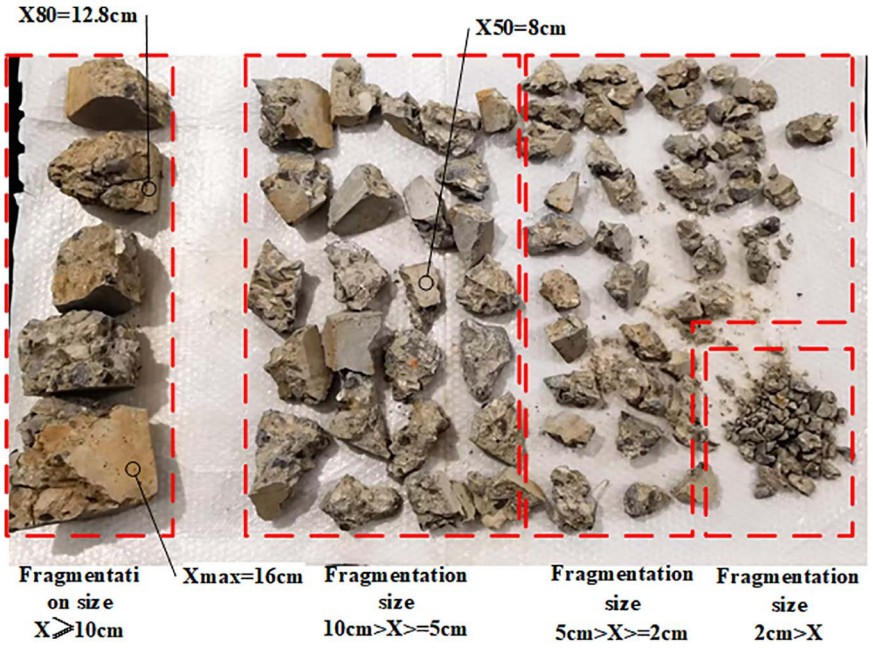

**Fig 3. Fragmentation statistics.**

**Table 4. Experimental fragmentation size for inter-hole delay.**

| Fragmentation size (cm) | Xmax | X80 | X50 | b | Swebrec fitted distribution function | R² |
|---|---|---|---|---|---|---|
| Group 1 | 16 | 12.8 | 8 | 1.32 | $P(x) = \dfrac{1}{1+[ln(16/x)/0.63]^{1.32}}$ | 0.978 |
| Group 2 | 16.2 | 13.5 | 9 | 1.18 | $P(x) = \dfrac{1}{1+[ln(16.2/x)/0.59]^{1.18}}$ | 0.999 |
| Group 3 | 19.8 | 17 | 9.2 | 0.83 | $P(x) = \dfrac{1}{1+[ln(19.8/x)/0.77]^{0.83}}$ | 0.991 |
| Group 4 | 14.5 | 12.8 | 7.2 | 0.78 | $P(x) = \dfrac{1}{1+[ln(14.5/x)/0.7]^{0.78}}$ | 0.992 |
| Group 5 | 19.4 | 12.6 | 7 | 1.6 | $P(x) = \dfrac{1}{1+[ln(19.4/x)/1.02]^{1.6}}$ | 0.999 |

fragmentation. Compared with simultaneous blasting, $X_{max}$, $X_{80}$, and $X_{50}$ were larger and less efficient in groups 2 and 3, while $X_{80}$ and $X_{50}$ were smaller and more effective in groups 4 and 5.

Plotting the Swebrec fitted distribution function shows that group 4 has a high fragmentation pass rate, more uniform size distribution, smaller average fragmentation size, and large fragmentation size, and fits the theoretical 3 ms optimal delay of the crushing effect.

### 4.2 Inter-row delay time data analysis

Statistics of the vibration data from the inter-row delay optimization experiment and plotting of the PPV and its rate of decrease are shown in Fig 5.

In the free face reverse, blasting vibration with blasting vibration between the rows of delay time and blast center distance increases and decreases, and the amplitude of the peak vibration velocity decreases with the increase in the delay time between the rows of increase with the change in the blast center distance shows a similar fluctuation pattern as the hole delay optimal experiment: the amplitude of the velocity drop with the increase in the blast center distance first decreases and then increases, and in the shorter delay time, this fluctuation pattern is more significant. Group 8 had the best damping effect, followed by group 4.

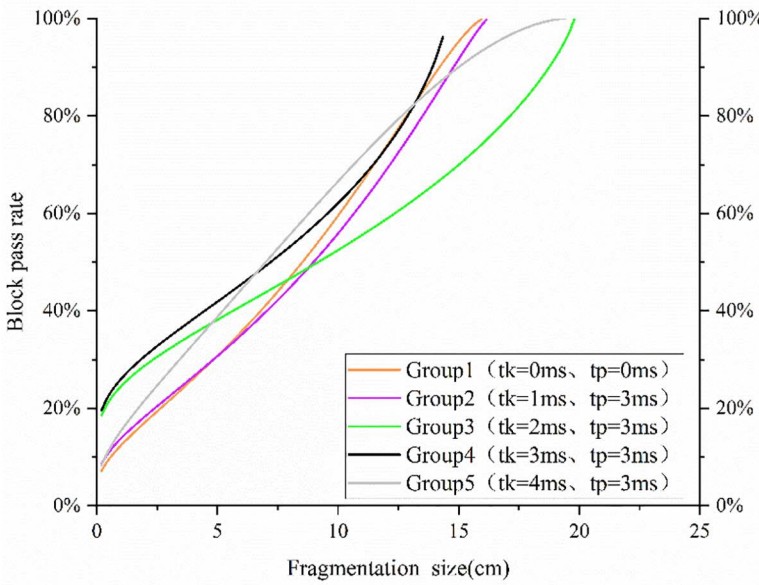

**Fig 4. Plot of Swebrec fitted distribution function.**

In the free face direction, the PPV with an increase in the delay time between rows exhibits an initial decrease followed by an increase. The best damping effect in this direction was observed in Group 7, followed by that in group 4.

In summary, groups 4, 7, and 8 are the combination of delay time with excellent damping effect: when the delay time between rows is 2 ms and 4 ms respectively, the damping effect is maximized in the free face direction and the free face is reversed; when the delay time between rows is 3 ms, the next level of damping effect can be realized in both directions.

Blasting vibration signals were analyzed in both time and frequency domains. Processed via MATLAB R2023a, the data adopted a 10 kHz sampling frequency and 5% Hanning window; dominant frequency was defined as the frequency corresponding to the maximum amplitude in the power spectrum. The change rule of the free face inverse principal frequency value is shown in Fig 6.

An observation of the distribution of the main frequency in the free-face reverse can be seen; with the increase in the burst center distance, the overall trend of attenuation, the main frequency of all delay programs fluctuates in the process of attenuation and cannot strictly follow the ideal attenuation law. In the bursting center distance smaller position, such as 0.5 m, the higher the frequency of the faster decay, blasting vibration main frequency with the increase in the delay time between rows and reduce the main frequency in the bursting center distance of 0.5 m to 1.7 m section of the main frequency of the regularity is not significant, and with the bursting center distance increased to 1.7 m, the main frequency of the experiments of the different rows of the delay time began to rapidly converge, which also shows that the effect of the delay time between the rows of the main frequency of the main frequency of the attenuation of the bursting center distance increases and Decreasing trend.

Notably, the main frequency of all optimized delay schemes (45-75 Hz, Fig 6) is much higher than the resonance frequency of common ground structures (1-10 Hz). The optimal 3 ms/3 ms (inter-hole/inter-row) scheme, for instance, achieves a main frequency of 58-72 Hz, avoiding resonance-induced vibration amplification and effectively reducing structural damage risks in blasting operations near surrounding structures. The other directions of the experiment also show a similar pattern of the main frequency, and in the region of the measurement point are maintained in the range of not resonating. The fragmentation size data is listed in the following Table 5.

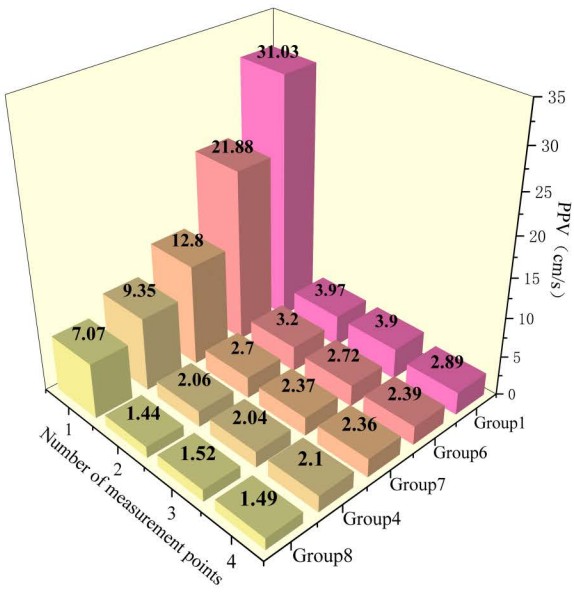

**(a).** Free face reverse PPV

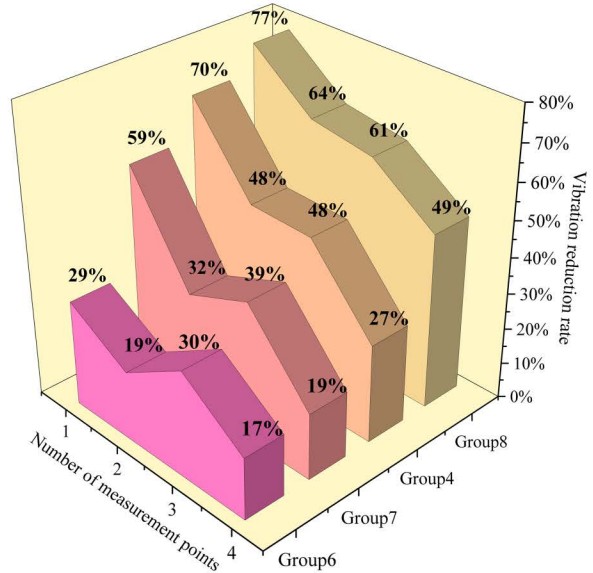

**(b).** Decrease rate of free face reverse PPV

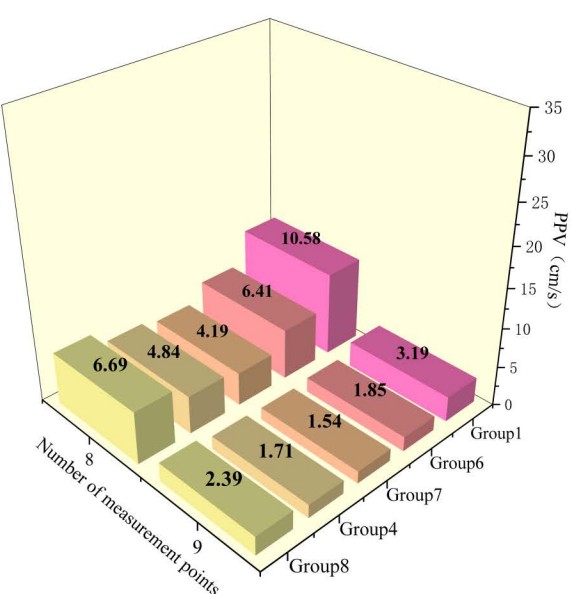

**(c).** Free face direction PPV

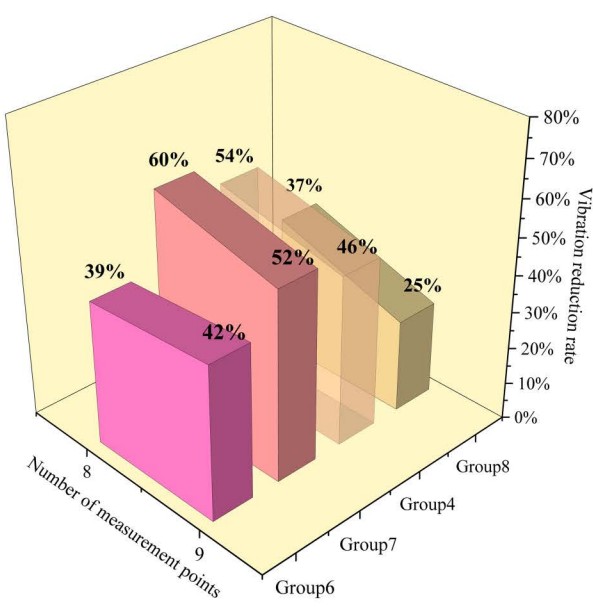

**(d).** Decrease rate of free face direction PPV

**Fig 5. Peak particle velocity (PPV) and vibration reduction rate of different inter-row delay schemes.** (a) Free face reverse PPV. (b) Decrease rate of free face reverse PPV. (c) Free face direction PPV. (d) Decrease rate of free face direction PPV.

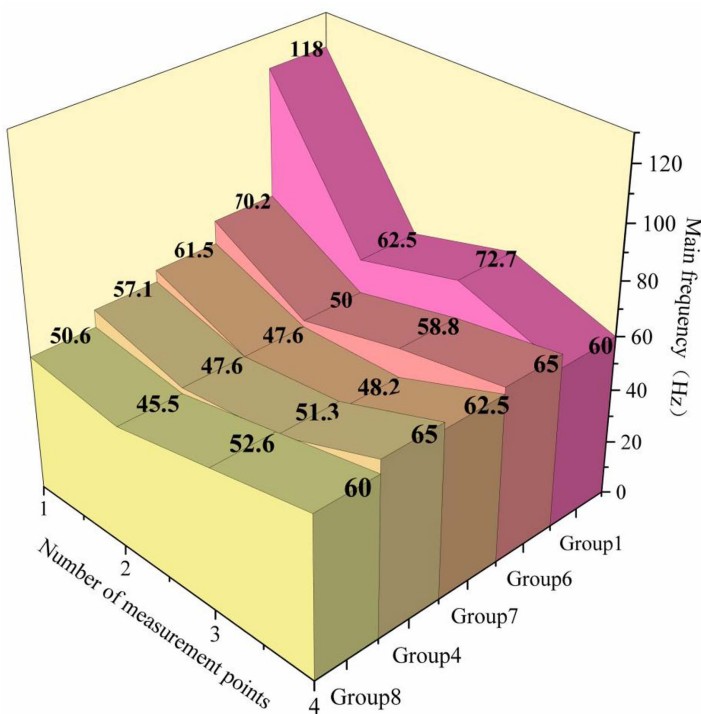

**Fig 6. Free face reverse principal frequency diagram.**

**Table 5. Experimental fragmentation size for inter-row delay.**

| Fragmentation size(cm) | $X_{max}$ | $X_{80}$ | $X_{50}$ | b | Swebrec fitted distribution function | $R^2$ |
|---|---|---|---|---|---|---|
| Group 1 | 16.0 | 12.8 | 8.0 | 1.32 | $P(x) = \frac{1}{1+[ln(16/x)/0.63]^{1.32}}$ | 0.978 |
| Group 4 | 14.5 | 12.8 | 7.2 | 0.78 | $P(x) = \frac{1}{1+[ln(14.5/x)/0.7]^{0.78}}$ | 0.992 |
| Group 6 | 15.0 | 12.4 | 7.8 | 1.12 | $P(x) = \frac{1}{1+[ln(15/x)/0.65]^{1.12}}$ | 0.988 |
| Group 7 | 15.5 | 15.0 | 9.7 | 0.52 | $P(x) = \frac{1}{1+[ln(15.5/x)/0.46]^{0.52}}$ | 0.993 |
| Group 8 | 15.8 | 13.0 | 8.5 | 1.18 | $P(x) = \frac{1}{1+[ln(15.8/x)/0.62]^{1.18}}$ | 0.999 |

Ouchterlony's Swebrec function fits a plot of the distribution function as shown in Fig 7.

A plot of the distribution function was fitted according to Swebrec, which clearly presents the distribution ratios of the rocks at different fragmentation sizes. Groups 4 and 7 had a higher percentage in the smaller fragmentation size range; however, the blasting effect of group 7 was significantly less effective, implying that the group produced more fragments of larger fragmentation sizes. Groups 4 and 6 outperformed the other groups in terms of larger fragmentation size pass rates, which indicates that, in these two groups, the maximum 50% of fragments were smaller in size relative to the other groups.

Based on the fragmentation size data, it can be seen that Group 4 and Group 6 successfully achieved a reduction in average fragmentation size compared to simultaneous blasting and the other groups. Groups 4 and 6 also reduced the fragmentation size in terms of symbolic fragmentation sizes $X_{80}$ and $X_{max.}$ With the increase in delay time between rows, the distribution of fragmentation size in different sizes exhibits an initial decrease, followed by an increase, then a

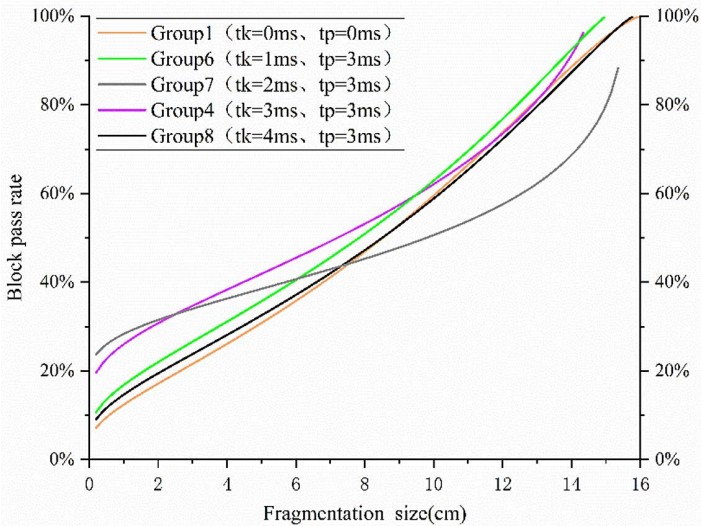

**Fig 7. Plot of Swebrec fitted distribution function.**

subsequent decrease and finally a further increase which reflects the complexity of the process of blasting debris generation and shows that it is not a simple linear correlation with the delay time.

In summary, group 4 and group 6 to achieve the optimization of the size distribution of the flush blast, that is, when the delay between the rows is set at 1 ms or 3 ms, the blasting effect is more ideal.

## 5. Field test validation

To further verify the reliability of the research methodology and core conclusions derived from laboratory-scale C50 concrete experiments, a field test was conducted on a real high-slope blasting project. The key objective is to confirm whether the optimization logic is applicable to complex engineering scenarios and to enhance the engineering applicability of this study's conclusions.

The test site is located on a 15-meter-high slope of a pumped storage power station, with jointed gneiss as the dominant rock mass and 14m-deep blast holes. The test site is shown in Fig 8. The sensors are deployed within a range of 17-80 meters from the blast center, symmetrically arranged along the direction of detonation propagation and its reverse direction to ensure the comprehensiveness of the data. The optimal delay combination was selected by using the delay time optimization method consistent with the laboratory experiment($t_k = 38\text{-}43$ ms, $t_p = 68\text{-}78$ ms).

### Field test results

The vibration reduction data are shown in Table 6.

Analysis of the field test data shows that the peak particle velocity (PPV) of both the original engineering group and the optimization group decreases with the increase of blast center distance, which is consistent with the attenuation law observed in laboratory experiments. Meanwhile, the vibration reduction rate of the optimized delay combination shows a gradual decreasing trend with the increase of blast center distance, with the highest reduction rate of 71.33% at the near-field position (20 m) and the lowest rate of 56.44% at the far-field position (35 m). This trend is highly consistent with the laboratory conclusion that "the vibration reduction effect is most significant in the near zone and gradually weakens in the far zone".

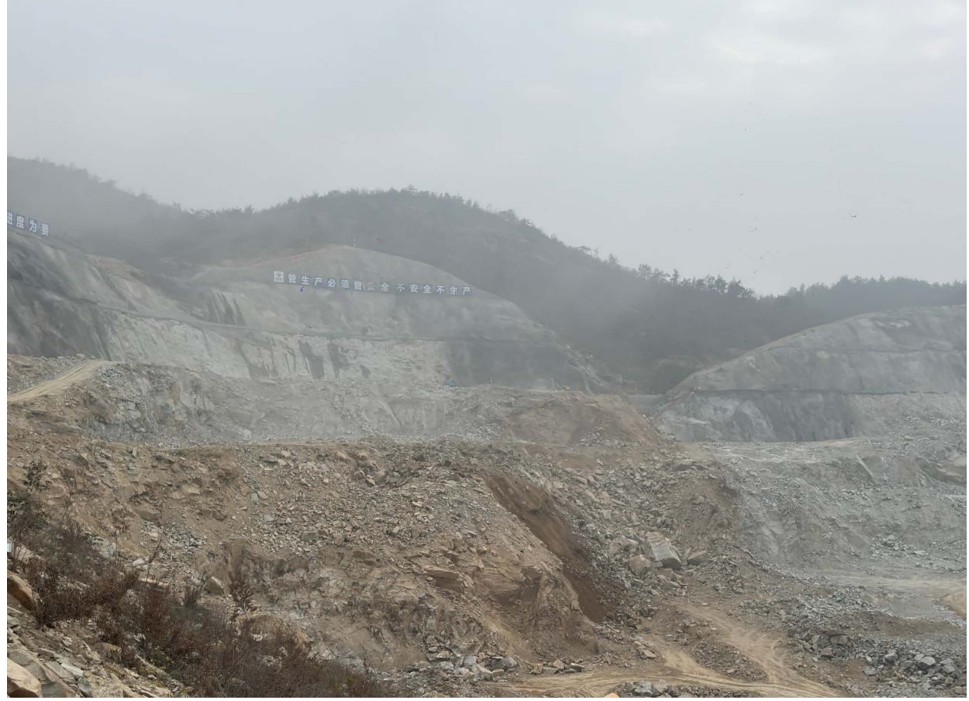

**Fig 8. On-site bench blasting test site of the high-slope project.**

Table 6. Vibration reduction rate of the delay time optimization groups ($t_k = 38\,ms$, $t_p = 68\,ms$) relative to the original engineering groups ($t_k = t_p = 13\,ms$).

| Bursting center distance（m） | Original engineering groups PPV(cm/s) ($t_k = t_p = 13\,ms$) | Optimization groups PPV(cm/s) ($t_k = 38\,ms$, $t_p = 68\,ms$) | Vibration reduction rate |
|---|---|---|---|
| 20 | 46.4 | 13.3 | 71.33% |
| 25 | 13.7 | 4.5 | 67.15% |
| 35 | 16.3 | 7.1 | 56.44% |

## 6. Discussion

This study has several limitations that need to be considered in engineering applications:

(1) The digital electronic detonators used in the laboratory have a metal shell design, which may exert additional local impact on the fragmentation of small-sized concrete specimens. Although this impact does not affect the screening of the optimal delay time combination, special attention should be paid to the difference in charge structure between the laboratory (using only detonators) and the field (using bulk explosives) when applying the research results to actual blasting projects.

(2) The experimental specimens are homogeneous C50 concrete without natural joints, which is essentially different from the jointed rock mass structure in the field. Therefore, for large-scale open-pit blasting projects, on-site parameter calibration must be performed.

## 7. Conclusion

This study establishes an interval optimization paradigm for delay-time selection through systematically controlled blasting experiments, achieving coupled control of vibration hazards and fragmentation efficiency. On the one hand, the structural damage or personnel safety hazards caused by the excessive PPV and the resonance phenomenon were successfully controlled; on the other hand, the rock crushing effect was further enhanced, the fragmentation size distribution was optimized, and the subsequent loading and transporting of fragments was more convenient and efficient, which effectively reduced the need for secondary crushing operations. The following conclusions are drawn from the experiments:

1. The study derived the theoretical optimal delay time value via the Hanukayev formula, and then determined the optimal delay time range ($t_k$ = 3-4 ms, $t_p$ = 1-3 ms) through comparative analysis of experimental PPV and blasting fragmentation data. The selection standard for this range is to compare all delay groups in parallel, identify the groups with the best and second-best vibration reduction and blasting effects respectively, and finally select the groups meeting both standards to delimit the optimal range. Specifically, the 3-4 ms inter-hole delay range corresponds to the interval with significantly lower PPV and higher fragmentation uniformity than other schemes, achieving a 54-70% reduction in PPV while maintaining 92% fragmentation uniformity.

2. The experiments revealed the sensitivity of PPV to the blasting direction and free surface conditions. Free face direction PPV decreases exponentially with increasing delay intervals, while free face reverse PPV follows linear scaling. Free face direction PPV of Group 4 ($t_k$ = 3 ms, $t_p$ = 3 ms) decreases exponentially from 10.01 cm/s (0.4 m blast center distance) to 2.14 cm/s (1.7 m blast center distance), conforming to the exponential law $y = 10.01e^{-0.8x}$ ($R^2$ = 0.96). Free face reverse PPV decreases linearly from 14.06 cm/s to 3.46 cm/s, following the linear law $y = -4.79x + 13.82$ ($R^2$ = 0.95).

3. Comprehensive experimental data clearly show that the optimal delay time is not an isolated fixed value, but an advantageous interval. In the interval of $t_k$ = 3-4 ms, $t_p$ = 1-3 ms, effective control of blasting vibration and optimization of rock fragmentation can be realized. Compared with traditional fixed-value delay design, the interval-based method has three core advantages:first, facilitates flexible selection of delay time within the interval by constructors based on actual conditions.second, ensures safe superimposed vibration intensity while avoiding poor rock fragmentation from excessive delays.third, adapts to changes in blast center distances and geological conditions, enhancing the applicability of the delay scheme.After rigorous comparative analysis, it was determined that the delay scheme with $t_k$ = 3 ms and $t_p$ = 3 ms is the comprehensive optimal solution, which significantly reduces the blasting vibration in the free face direction and its reverse direction by 70% and 54%, respectively, and reduces the average and large sizes of rock fragments by 10% and 9.3%, respectively, which has certain guiding significance in the practice of blasting engineering. However, considering the complexity of actual engineering, it is still necessary to adjust the parameters to a certain extent according to the actual situation on site when applying this preferred method of delay time to blasting engineering.

4. A complementary field test on a 15 m-high slope further validates the study's reliability. Adopting the same interval-based delay optimization framework as the laboratory experiment, the field test determined the initial delay interval via Hanukayev's theory and jointed gneiss mechanical parameters, screening the optimal combination ($t_k$ = 38 ms, $t_p$ = 68 ms). Despite scale-effect-induced differences in absolute delay values (vs. laboratory's $t_k$ = 3-4 ms, $t_p$ = 1-3 ms), the core mechanism of vibration reduction through rational stress wave interference remains consistent-achieving a maximum vibration reduction rate of 71.33% at 20 m blast center distance. This confirms the laboratory-derived method can be effectively migrated to large-scale open-pit blasting of jointed rock masses, verifying its reliability and engineering applicability.

## Supporting information

**S1 Data. Experimental data.**

(ZIP)

## Author contributions

**Conceptualization:** Li He, Sheng Peng.

**Data curation:** Li He, Wuyi Zhang, Yongming Zhao, Maolin Wang.

**Formal analysis:** Wuyi Zhang, Yongming Zhao, Shasha Chen, Maolin Wang.

**Investigation:** Shasha Chen.

**Methodology:** Yongming Zhao.

**Resources:** Li He, Maolin Wang.

**Software:** Wuyi Zhang.

**Supervision:** Li He, Sheng Peng.

**Validation:** Yongming Zhao, Yingkang Yao, Shasha Chen.

**Visualization:** Yingkang Yao.

**Writing – original draft:** Wuyi Zhang.

**Writing – review & editing:** Li He, Sheng Peng.

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
