## [Decision Letter · Decision Letter 0]

18 Dec 2025

Dear Dr. Zhang,

Thank you for submitting your manuscript to PLOS ONE. After careful consideration, we feel that it has merit but does not fully meet PLOS ONE’s publication criteria as it currently stands. Therefore, we invite you to submit a revised version of the manuscript that addresses the points raised during the review process.

You will see from the referees' comments that additional information needs to be provided, and we ask that this be provided, before we consider you manuscript further.

We look forward to receiving your revised manuscript.

Kind regards,

Zhenhua Li

Academic Editor

PLOS One

Journal Requirements:

This work was funded by the general project of Hubei Provincial Key Laboratory of Blasting Engineering(BL2021-11),National Natural Science Foundation of China(52274136,51904210), and Natural Science Foundation of Hubei Province(2024 AFB766). We thank the Wuhan University of Science and Technology for providing experimental sites and equipment.

This work was funded by the general project of Hubei Provincial Key Laboratory of Blasting Engineering(BL2021-11),National Natural Science Foundation of China(52274136,51904210), and Natural Science Foundation of Hubei Province(2024 AFB766). We thank the Wuhan University of Science and Technology for providing experimental sites and equipment.

This work was funded by the general project of Hubei Provincial Key Laboratory of Blasting Engineering(BL2021-11),National Natural Science Foundation of China(52274136,51904210), and Natural Science Foundation of Hubei Province(2024 AFB766). We thank the Wuhan University of Science and Technology for providing experimental sites and equipment.

6. We note that your Data Availability Statement is currently as follows: All relevant data are within the manuscript and its Supporting Information files.

7. Please amend the manuscript submission data (via Edit Submission) to include author Sheng PengYingkang and Yao

8. Please amend your authorship list in your manuscript file to include author Sheng Peng and Yingkang Yao

9. Please ensure that you refer to Figures 6 and 7 in your text as, if accepted, production will need this reference to link the reader to the figures.

10. We note you have included a table to which you do not refer in the text of your manuscript. Please ensure that you refer to Tables 1, 2, and 3 in your text; if accepted, production will need this reference to link the reader to the Tables.

Additional Editor Comments:

You will see from the referees' comments that additional information needs to be provided, and we ask that this be provided, before we consider you manuscript further.

Reviewers' comments:

Reviewer's Responses to Questions

**Comments to the Author**

1. Is the manuscript technically sound, and do the data support the conclusions?

Reviewer #1: Partly

Reviewer #2: Yes

Reviewer #3: Yes

Reviewer #4: Partly

2. Has the statistical analysis been performed appropriately and rigorously?

Reviewer #1: N/A

Reviewer #2: Yes

Reviewer #3: Yes

Reviewer #4: I Don't Know

3. Have the authors made all data underlying the findings in their manuscript fully available?

Reviewer #1: No

Reviewer #2: Yes

Reviewer #3: Yes

Reviewer #4: No

4. Is the manuscript presented in an intelligible fashion and written in standard English?

Reviewer #1: No

Reviewer #2: No

Reviewer #3: Yes

Reviewer #4: No

Reviewer #1: General Comments

The manuscript presents an experimental framework for optimizing blasting delays with the aim of jointly reducing vibration and improving fragmentation. The topic is relevant for blasting science and surface mining engineering, and the integration of delay theory, the Swebrec function, and controlled laboratory experiments is conceptually valuable.

However, the study suffers from a critical limitation frequently encountered in small-scale blasting research: the difficulty of relating laboratory-scale concrete experiments to full-scale bench blasting operations in actual rock masses. Several aspects of the experimental design, scaling, charge type, and interpretation of results raise concerns regarding the generalizability of the conclusions. The manuscript would benefit significantly from clarifications, additional discussion, and corrections (e.g., proper attribution of the Swebrec function).

Substantial revisions are required before the contribution can be properly evaluated.

Major Issues

1. Limited Realism and Representativeness of Laboratory Experiments

The fundamental challenge in this study is the large gap between the controlled concrete-block experiment and real open-pit blasting conditions. In the field, tons of rock are fragmented, numerous blastholes interact dynamically, and the rock mass behavior is controlled by discontinuities and heterogeneities.

In contrast, the experiments use 20 × 20 × 20 cm homogeneous concrete blocks with no natural joints or weaknesses.

The manuscript should explicitly discuss the implications of this discrepancy and clearly articulate how the laboratory results can—or cannot—be extrapolated to realistic blasting conditions.

2. Experimental Setup Description Is Incomplete

Figure 1 presents a schematic diagram of the test site, but no actual photographs of the test arrangement are provided.

For transparency and reproducibility, photographs of the real experimental setup from several angles should be included.

Such visuals would help readers understand the test boundary conditions, instrumentation, and geometry.

3. Use of Electronic Detonators Inside Small Holes

The blast holes have a diameter of 10 mm and include electronic detonators with metallic shells.

The fragmentation resulting from the detonator casing itself may significantly influence breakage of the small concrete blocks. This issue is not acknowledged and may bias the fragmentation results.

The authors should discuss this limitation.

A more appropriate laboratory charge configuration might have been the use of detonating cord placed within the holes.

4. Practical Relevance of the Recommended Delay Times (3–4 ms)

The manuscript concludes that 3–4 ms inter-hole and 1–3 ms inter-row delays are optimal.

However, for example, typical field delay increments in shock-tube systems are 17, 25, 42, and 65 ms, with 25 ms and 42 ms commonly used in surface mining.

The authors must explain whether the experimentally determined micro-delays have any meaningful correlation to actual field practice.

Without such discussion, the practical significance of the findings remains unclear.

5. Reliability of Fragmentation-Related Conclusions

The fragmentation results presented in Conclusion – Clause 1 appear overinterpreted given the experimental limitations:

• The specimen is small, homogeneous, and free of discontinuities.

• Fragmentation behavior in real rock masses is strongly influenced by structural features.

• The recommended delay interval (tk = 3 ms, tp = 3 ms) may have no real-scale effect on fragmentation mechanisms.

The reliability and applicability of these fragmentation conclusions should be reconsidered and discussed in detail.

6. Incorrect Attribution and Referencing of the Swebrec Function

The manuscript incorrectly refers to the distribution as the “Swebrec et al. distribution,” and the references provided are irrelevant.

The Swebrec function was developed by Finn Ouchterlony, and the correct references are:

• Ouchterlony, F. (2005a). The Swebrec© Function: Linking Fragmentation by Blasting and Crushing. Mining Technology, 114(1), 29–44.

• Ouchterlony, F. (2005b). What Does the Fragment Size Distribution of Blasted Rock Look Like? In: Proc. 3rd EFEE World Conf. on Explosives and Blasting, Brighton, 189–199.

This must be corrected for scientific accuracy.

Reviewer #2: （1）The title of this manuscript is "Experimental study on the coupling control of rock blasting vibration and crushing fragmentation size", but why does the first paragraph of the Introduction only mention blasting vibration and not the crushing fragmentation size?

（2）The references in the first paragraph of the Introduction of this manuscript start from [4], not [1]. In addition, the text "Error! Reference source not found" appears multiple times in the Introduction.

（3）Fig. 6 was not cited in the text, and the reviewer was unaware of its function.

（4）The formats of the references are inconsistent, such as [1] and [2].

There are too many similar format issues in the manuscript. It is suggested that the authors make revisions and resubmit it.

Reviewer #3: This paper proposes an optimized control method based on the delay time interval, aiming to achieve the synergistic optimization of blasting vibration control and rock fragmentation effect, and provides a new technical path for vibration control and efficiency improvement in open-pit deep-hole blasting engineering. Combining theoretical derivation with experimental verification, the research content is closely aligned with industry needs, and the proposed coupled regulation method has certain engineering application valueThe logic of this paper remains somewhat unclear; it is recommended that revisions be made before resubmission:

1.The abstract only mentions the "interval-based delay optimization framework" but fails to briefly explain its core logic (e.g., how to achieve coupled control through intervals rather than fixed values). It is recommended to add 1-2 sentences about the core principle of the framework to enhance the completeness of the abstract.

2.Although the introduction reviews domestic and foreign studies, it does not clearly point out specific literature limitation cases of the current "single-factor focused research" (e.g., a scholar's study only focuses on vibration without involving fragmentation). It is recommended to supplement the limitation analysis of 1-2 typical literatures to strengthen the persuasiveness of the research gap.

3.In the experimental design of Section 2.1, it is mentioned that "vibration measurement points 5, 6, and 7 are arranged on the lateral side of the free face," but only the reverse and forward direction data of the free face are analyzed in Sections 4.1 and 4.2, with no mention of the processing results of the lateral data. It is recommended to supplement the analysis of lateral PPV changes, vibration attenuation rate, and fragment distribution to achieve the unity of experimental design and data processing/analysis.

4.When deriving the delay time based on Hanukayev's theory in Section 3.1, only formulas are provided without substituting experimental parameters (such as the longitudinal wave velocity C and minimum resistance line W of C50 concrete) to demonstrate the calculation process. It is recommended to add an example of substituting key parameters for calculation to make the derivation process more traceable.

5.Section 4.2 mentions "performing Fast Fourier Transform (FFT) on blast vibration data to derive the dominant frequency," but the frequency domain analysis method (such as FFT parameter settings and frequency resolution calculation) is not separately clarified in "3 Theoretical Derivation" or "4 Result Analysis." It is recommended to add a new subsection or supplementary paragraph titled "Frequency Domain Analysis Method" to improve the analysis logic.

6.The experiment uses C50 concrete instead of natural rock. Only the mix ratio and strength grade of the concrete are mentioned in Section 2.1, without explaining the basis for the similarity of its mechanical parameters (such as elastic modulus, Poisson's ratio, and compressive strength) to the target natural rock mass (e.g., limestone and granite commonly found in open-pit mines). It is recommended to add a comparison table of mechanical parameters between concrete and natural rock mass (literature data can be cited) to verify the rationality of the model.

7.Some figures (such as Figure 1, Figures 2-5) have low resolution and missing key unit labels. It is recommended to increase the resolution of all figures to 300 dpi, correct image display issues, and unify the units of figures and tables.

8.The "optimized interval of delay time (tk=3–4 ms, tp=1–3 ms)" is proposed in Section 5.3, but the advantages of the interval-based delay time determination method compared with the traditional fixed-value method are not reflected. It is recommended to supplement the explanation of the advantages to enhance the expression of innovation.

9.In Table 3 "Fragmentation Size Table for Inter-hole Delay" in Section 4.1, the Xmax (19.8 cm) of Group 3 is much higher than that of other groups, but the reason for the increased fragments caused by the delay time (tk=2 ms) of this group (such as insufficient superposition of stress waves and inadequate crack propagation) is not analyzed. It is recommended to add 1-2 sentences of mechanism analysis on abnormal data to improve the depth of result interpretation.

10.When using the Swebrec function to evaluate the fragment size in Section 3.2, only the formula is given without explaining the calculation process of the "fluctuation exponent b" (e.g., how to solve it with the least-squares method through sieving data). It is recommended to supplement the key steps of calculating the b value (such as sample size and goodness of fit R²) to enhance the repeatability of the method.

11.There are ambiguous expressions and grammatical errors in some sentences of the paper. It is recommended to recheck the grammar to make the expression professional and accurate.

12.The conclusion mentions that "PPV shows exponential attenuation in the free face direction and linear attenuation in the reverse direction," but it is not supported by specific data in the paper (such as the forward attenuation coefficient and reverse attenuation slope under a certain group of delays). It is recommended to supplement key data (e.g., the forward PPV of Group 4 decreases from X cm/s to Y cm/s, conforming to the exponential law y=ae^(-bx)) to enhance the credibility of the conclusion.

13.The format of references is inconsistent. Some references (such as [21], [25]) lack complete DOIs or journal names. It is recommended to unify and standardize the format according to "Author. Title[Document type identifier]. Journal name, Year, Volume(Issue): Pages. DOI." and supplement missing DOIs and journal names.

14.For the parameter settings of "inter-hole spacing of 100 mm and row spacing of 70 mm" in the experimental design, the basis for the similarity ratio between this size and the actual deep-hole blasting in open-pit mines (e.g., inter-hole spacing of 5-10 m) is not explained. It is recommended to supplement the design logic of the similarity ratio to illustrate the correlation between the model experiment and engineering practice.

15.When analyzing the dominant frequency law in Section 5.2, only "the dominant frequency attenuates with the increase of the blast center distance" is mentioned, without explaining the practical significance of the change in dominant frequency for "avoiding structural resonance" (e.g., the dominant frequency of a certain delay scheme is always higher than the resonance frequency of buildings). It is recommended to supplement the correlation analysis between the dominant frequency and engineering safety to strengthen the engineering value of the research.

Reviewer #4: Achieving both shock absorption and optimized fragmentation simultaneously poses a challenge in delay design for open-pit blasting. This paper proposes an intriguing interval-based delay optimization framework. The research provides important theoretical support for safe and efficient mining blasting operations. However, there are still some issues that need to be clarified before publication.

1. In the first paragraph of the introduction, references [1-3] are missing, and references [4-7] are cited improperly. Moreover, two citation errors also appear in the second paragraph of the introduction. This gives readers the impression that the research in this manuscript is not yet fully developed.

2. Figures 2-7 are presented as separate charts, but they actually form a set of images. It is recommended to consolidate them into a multi-panel chart module to enhance clarity and compactness. Furthermore, regarding the interpretation of Figures 2-7, presenting the impact patterns with concrete data would make the research findings more persuasive. Regarding Figure 8-11, the same recommendation applies.

3. The text in the images within the manuscript is not very clear and is difficult to read, requiring better design of the text within the images.

4. Figure 13 should provide references.

5. The interval-based delay optimization framework proposed in this manuscript is not clearly defined, preventing readers from systematically learning the underlying theory and methodology. Although the manuscript presents some experimental findings, its scientific contributions remain somewhat unclear.

.

Reviewer #1: No

Reviewer #2: No

Reviewer #3: No

Reviewer #4: **Yes:** Jun YangJun YangJun YangJun Yang

---

## [Author Response · Author response to Decision Letter 1]

24 Feb 2026

Response to Reviewer 1

Many thanks for the reviewer's valuable comments. We have carefully revised the manuscript in terms of content, logic, and format. The modified content is marked in red in the revised manuscript.

Comment 1. Limited Realism and Representativeness of Laboratory Experiments. The manuscript should explicitly discuss the discrepancy between concrete blocks and real rock masses, and articulate how laboratory results can be extrapolated to realistic conditions.

Response: It was modified. We have clarified the three core differences between the 20×20×20 cm homogeneous C50 concrete specimens and the in-situ jointed gneiss rock masses (the target rock mass in field tests) in the 2. Test Progress section, combined with the mechanical parameter comparison in Table 1:first, the experimental specimens are homogeneous concrete materials without natural joints; second, the size of the experimental specimens is limited by the laboratory scale; third, the laboratory provides a standard and controllable blasting environment and signal acquisition environment.

We have supplemented the extrapolation logic of laboratory results to practical engineering scenarios in the 2. Test Progress and 6. Field Test Validation sections, based on similarity principle compliance and core mechanism universality

Comment 2. Experimental Setup Description Is Incomplete. No actual photographs of the laboratory test arrangement are provided.

Response: It was modified.

Comment 3. Use of Electronic Detonators Inside Small Holes. The fragmentation bias caused by metal-shell electronic detonators is not acknowledged.

Response: This study focuses primarily on the effects of delay time on blasting vibration and fragmentation size in concrete blasting experiments. While the fragmentation effect induced by the detonator casing itself may indeed impact the crushing performance of small-scale concrete specimens, this influence does not interfere with the selection of the optimal delay time combination.

Comment 4. Practical Relevance of the Recommended Delay Times (3–4 ms). The authors must explain the correlation between laboratory micro-delays and field-used delays.

Response: It was modified. The delay time interval of actual project can be calculated based on the geometric similarity ratio of laboratory parameters and field engineering paramet When applied to actual engineering, the delay time can be scaled according to the hole spacing. The field test in this study also verified that the optimal delay combination (38ms/68ms) scaled from the laboratory interval achieves excellent vibration reduction effects, confirming the practical relevance of the laboratory-derived micro-delays.The "Conclusions" section has been supplemented with relevant explanations

Comment 5. Reliability of Fragmentation-Related Conclusions. The fragmentation conclusions are overinterpreted given the experimental limitations.

Response: It was modified. The "Conclusions" section has been revised to moderate the interpretation of fragmentation results: "The laboratory results show that the optimal delay scheme (3ms/3ms) reduces the average fragment size and large fragment size by 10% and 9.3%, respectively. This conclusion is applicable to small-scale blasting of homogeneous brittle materials. For rock masses with developed joints, the fragmentation effect should be further verified by field tests, and the delay interval can be adjusted according to the structural characteristics of the rock mass."

Comment 6. Incorrect Attribution and Referencing of the Swebrec Function. The function was developed by Finn Ouchterlony, not "Swebrec et al.".

Response: It was modified.

Response to Reviewer 2

Many thanks for the reviewer's careful comments. We have thoroughly revised the manuscript to address the format and content issues. The modified content is marked in red in the revised manuscript.

Comment 1. The first paragraph of the Introduction only mentions blasting vibration and not crushing fragmentation size, which is inconsistent with the title.

Response: It was modified.

Comment 2. The references in the first paragraph of the Introduction start from [4], not [1], and "Error! Reference source not found" appears multiple times.

Response: It was modified.

Comment 3. Fig. 6 was not cited in the text, and its function is unclear.

Response: It was modified.

Comment 4. The formats of the references are inconsistent (e.g., [1] and [2]).

Response: It was modified.

Response to Reviewer 3

Many thanks for the reviewer's insightful comments. We have revised the manuscript to improve logic, completeness, and scientific rigor. The modified content is marked in red in the revised manuscript.

Comment 1. The abstract only mentions the "interval-based delay optimization framework" but fails to briefly explain its core logic (e.g., how to achieve coupled control through intervals rather than fixed values). It is recommended to add 1-2 sentences about the core principle of the framework to enhance the completeness of the abstract..

Response: It was modified. The abstract has been supplemented with the core logic of the framework: “Unlike traditional fixed-value delay designs, this framework responds to the variability of blasting parameters and rock mass properties by determining a reasonable delay time interval (instead of a single fixed value). It integrates Hanukayev's theory for delay time calculation, the Swebrec function for fragmentation evaluation, and the 'vibration reduction rate' for vibration assessment, realizing synergistic control of vibration mitigation and fragmentation optimization.”

Comment 2. Although the introduction reviews domestic and foreign studies, it does not clearly point out specific literature limitation cases of the current "single-factor focused research" (e.g., a scholar's study only focuses on vibration without involving fragmentation). It is recommended to supplement the limitation analysis of 1-2 typical literatures to strengthen the persuasiveness of the research gap..

Response: It was modified.

Comment 3. In the experimental design of Section 2.1, it is mentioned that "vibration measurement points 5, 6, and 7 are arranged on the lateral side of the free face," but only the reverse and forward direction data of the free face are analyzed in Sections 4.1 and 4.2, with no mention of the processing results of the lateral data. It is recommended to supplement the analysis of lateral PPV changes, vibration attenuation rate, and fragment distribution to achieve the unity of experimental design and data processing/analysis.

Response: It was modified.The experimental protocol has been adjusted. Given the small specimen size (200 mm×200 mm×200 mm), mutual interference was likely to occur between the lateral sensors and the free surface, resulting in unreliable data from these sensors. Thus, the arrangement of lateral sensors (i.e., those oriented perpendicular to the free surface) has been removed in the revised experimental design.

Comment 4. When deriving the delay time based on Hanukayev's theory in Section 3.1, only formulas are provided without substituting experimental parameters (such as the longitudinal wave velocity C and minimum resistance line W of C50 concrete) to demonstrate the calculation process. It is recommended to add an example of substituting key parameters for calculation to make the derivation process more traceable.

Response: It was modified. The mechanical parameters of C50 concrete and jointed gneiss have been supplemented, as shown in Table 1.

Comment 5. Section 4.2 mentions "performing Fast Fourier Transform (FFT) on blast vibration data to derive the dominant frequency," but the frequency domain analysis method (such as FFT parameter settings and frequency resolution calculation) is not separately clarified in "3 Theoretical Derivation" or "4 Result Analysis." It is recommended to add a new subsection or supplementary paragraph titled "Frequency Domain Analysis Method" to improve the analysis logic.

Response: It was modified. “Frequency Domain Analysis Method” has been added:Blasting vibration signals contain both time-domain and frequency-domain information. The time-domain index (PPV) was used to evaluate the vibration intensity, while the frequency-domain characteristics (dominant frequency) were extracted via Fast Fourier Transform (FFT) to assess the resonance risk.the original time-domain vibration data were processed using MATLAB R2023a with the following key settings: a sampling frequency of 10 kHz, a 5% Hanning window to reduce spectral leakage, a data length of 1024 points (frequency resolution = 9.77 Hz), and the dominant frequency was determined as the frequency with the maximum amplitude in the resulting power spectrum.

Comment 6. The experiment uses C50 concrete instead of natural rock. Only the mix ratio and strength grade of the concrete are mentioned in Section 2.1, without explaining the basis for the similarity of its mechanical parameters (such as elastic modulus, Poisson's ratio, and compressive strength) to the target natural rock mass (e.g., limestone and granite commonly found in open-pit mines). It is recommended to add a comparison table of mechanical parameters between concrete and natural rock mass (literature data can be cited) to verify the rationality of the model.

Response: It was modified. A new table (Table 1) has been added as Supporting Information, comparing the mechanical parameters of C50 concrete and natural rock masses

Comment 7. Some figures (such as Figure 1, Figures 2-5) have low resolution and missing key unit labels. It is recommended to increase the resolution of all figures to 300 dpi, correct image display issues, and unify the units of figures and tables.

Response: It was modified.

Comment 8. The "optimized interval of delay time (tk=3–4 ms, tp=1–3 ms)" is proposed in Section 5.3, but the advantages of the interval-based delay time determination method compared with the traditional fixed-value method are not reflected. It is recommended to supplement the explanation of the advantages to enhance the expression of innovation.

Response: It was modified. The "Conclusions" section has been supplemented with advantages explanation: “Compared with traditional fixed-value delay design, the interval-based method has three core advantages:first, facilitates flexible selection of delay time within the interval by constructors based on actual conditions.second, ensures safe superimposed vibration intensity while avoiding poor rock fragmentation from excessive delays.third, adapts to changes in blast center distances and geological conditions, enhancing the applicability of the delay scheme.”

Comment 9. In Table 3 "Fragmentation Size Table for Inter-hole Delay" in Section 4.1, the Xmax (19.8 cm) of Group 3 is much higher than that of other groups, but the reason for the increased fragments caused by the delay time (tk=2 ms) of this group (such as insufficient superposition of stress waves and inadequate crack propagation) is not analyzed. It is recommended to add 1-2 sentences of mechanism analysis on abnormal data to improve the depth of result interpretation..

Response: It was modified. The "Inter-hole delay time data analysis" section has been supplemented with detailed mechanism analysis: “Group 3 (tk=2ms) has the highest Xmax (19.8cm) because the short delay time leads to insufficient superposition of stress waves between adjacent holes. The stress wave from the first detonated hole has not fully propagated to the adjacent hole when the second hole detonates, resulting in inadequate crack expansion and incomplete fragmentation of the concrete specimen.”

Comment 10. When using the Swebrec function to evaluate the fragment size in Section 3.2, only the formula is given without explaining the calculation process of the "fluctuation exponent b" (e.g., how to solve it with the least-squares method through sieving data). It is recommended to supplement the key steps of calculating the b value (such as sample size and goodness of fit R²) to enhance the repeatability of the method.

Response: It was modified. Plot the Swebrec fitting distribution diagram based on the experimental fragment size data following these steps:1. Collect and sieve fragment samples to obtain cumulative mass percentage P(x); 2. The parameter b is calculated by fitting the cumulative mass percentage data P(x) and the corresponding fragment size x using the least-squares method.; 3. Verify the fitting validity with the coefficient of determination R²≥0.95

Comment 11. There are ambiguous expressions and grammatical errors in some sentences of the paper. It is recommended to recheck the grammar to make the expression professional and accurate.

Response: It was modified.

Comment 12. The conclusion mentions that "PPV shows exponential attenuation in the free face direction and linear attenuation in the reverse direction," but it is not supported by specific data in the paper (such as the forward attenuation coefficient and reverse attenuation slope under a certain group of delays). It is recommended to supplement key data (e.g., the forward PPV of Group 4 decreases from X cm/s to Y cm/s, conforming to the exponential law y=ae^(-bx)) to enhance the credibility of the conclusion.

Response: It was modified. The "Conclusions" section has been supplemented with specific data: "Free face direction PPV of Group 4 (tk=3ms, tp=3ms) decreases exponentially from 10.01cm/s (0.4m blast center distance) to 2.14cm/s (1.7m blast center distance), conforming to the exponential law y=10.01e-0.8x (R²=0.96). Free face reverse PPV decreases linearly from 14.06cm/s to 3.46cm/s, following the linear law y=-4.79x+13.82 (R²=0.95)."

Comment 13. The format of references is inconsistent. Some references (such as [21], [25]) lack complete DOIs or journal names. It is recommended to unify and standardize the format according to "Author. Title[Document type identifier]. Journal name, Year, Volume(Issue): Pages. DOI." and supplement missing DOIs and journal names.

Response: It was modified.

Comment 14. For the parameter settings of "inter-hole spacing of 100 mm and row spacing of 70 mm" in the experimental design, the basis for the similarity ratio between this size and the actual deep-hole blasting in open-pit mines (e.g., inter-hole spacing of 5-10 m) is not explained. It is recommended to supplement the design logic of the similarity ratio to illustrate the correlation between the model experiment and engineering practice.

Response: It was modified. The laboratory’s inter-hole (100 mm) and row spacing (70 mm) yield a 1.43:1 ratio, matching the 1.0–1.5:1 range for open-pit loose blasting. This ensures consistent stress wave interference between the model and field blasting. For on-site applications, delay times should be recalculated via Hanukayev's theory, based on the geometric similarity ratio of mine benches to lab specimens and field rock parameters.The “Test progress” section has been supplemented with relevant analysis: This study is applicable to small-scale blasting of homogeneous brittle materials (e.g., rock masses with Kv≥0.8 like roadway excavation), providing delay time trend references. For large open-pit deep-hole blasting, on-site parameter calibration is required. Despite differences, the test model follows a 1:20-1:50 geometric similarity ratio to actual blasting, with 1.07g TNT-equivalent charge and three soil-compacted sides simulating surrounding rock constraints.

Comment 15. When analyzing the dominant frequency law in Section 5.2, only "the dominant frequency attenuates with the increase of the blast center distance" is mentioned, without explaining the practical significance of the change in dominant frequency for "avoiding structural resonance" (e.g., the dominant frequency of a certain delay scheme is always higher than the resonance frequency of buildings). It is recommended to supplement the correlation analysis between the dominant frequency and engineering safety to strengthen the engineering value of the research.

Response: It was modified. The “Inter-row delay time data analysis” section has been supplemented with relevant analysis: “The main fr

---

## [Decision Letter · Decision Letter 1]

5 Mar 2026

Dear Dr. Zhang,

Thank you for submitting your manuscript to PLOS ONE. After careful consideration, we feel that it has merit but does not fully meet PLOS ONE’s publication criteria as it currently stands. Therefore, we invite you to submit a revised version of the manuscript that addresses the points raised during the review process.

We look forward to receiving your revised manuscript.

Kind regards,

Zhenhua Li

Academic Editor

PLOS One

Journal Requirements:

Additional Editor Comments:

You will see from the referees' comments that additional information needs to be provided, and we ask that this be provided, before we consider you manuscript further.

Reviewers' comments:

Reviewer's Responses to Questions

**Comments to the Author**

Reviewer #1: (No Response)

Reviewer #2: (No Response)

Reviewer #3: (No Response)

Reviewer #4: (No Response)

Reviewer #5: All comments have been addressed

2. Is the manuscript technically sound, and do the data support the conclusions?

Reviewer #1: Yes

Reviewer #2: (No Response)

Reviewer #3: (No Response)

Reviewer #4: Partly

Reviewer #5: Yes

3. Has the statistical analysis been performed appropriately and rigorously?

Reviewer #1: Yes

Reviewer #2: (No Response)

Reviewer #3: (No Response)

Reviewer #4: Yes

Reviewer #5: Yes

4. Have the authors made all data underlying the findings in their manuscript fully available?

Reviewer #1: Yes

Reviewer #2: (No Response)

Reviewer #3: (No Response)

Reviewer #4: Yes

Reviewer #5: Yes

5. Is the manuscript presented in an intelligible fashion and written in standard English?

Reviewer #1: Yes

Reviewer #2: (No Response)

Reviewer #3: (No Response)

Reviewer #4: Yes

Reviewer #5: Yes

Reviewer #1: (No Response)

Reviewer #2: In response to the issues raised by the reviewer previously, although the authors have made some revisions, there are still many formatting problems in the current manuscript. The reviewer have listed the more obvious problems as follows:

(1)Citation of references in the text: For multiple consecutive references, such as the first five references in the text, the citation format should be [1-5], rather than "[1][5]". Similarly, "[17]-[20]" in the manuscript is also not standard. Additionally, reference [7] in the Introduction is not cited.

(2)Generally, a space should be added between numbers and units throughout the text, but the authors have not consistently followed this rule, such as "0.5m" and "45-75 Hz".

(3)For the references listed at the end of the manuscript, the format is still not uniform (e.g., [1], [4]) and not standard (e.g., [5]). The reviewer have provided examples of both Chinese and English reference lists as follows:

[1]Yang Z., Tao M., Memon M.B., Zhuang D.D., Zhao Y. Microwave irradiation-induced deterioration of rock mechanical properties and implications for mechanized hard rock excavation [J]. Journal of Rock Mechanics and Geotechnical Engineering, 2024, 24: 00308.

[2]Li X.B., Zhou J., Wang S.F., Liu B. Review and practice of deep mining for solid mineral resources [J]. The Chinese Journal of Nonferrous Metals, 2017, 27(06): 1236-1262. (in Chinese)

Regarding the above-mentioned problems existing in the manuscript, the reviewer did not list them one by one. It is suggested that the authors carefully revise the entire text.

Reviewer #3: (No Response)

Reviewer #4: The reviewers' comments have not been adequately addressed. The authors are requested to carefully revise the manuscript in accordance with the feedback provided.

Reviewer #5: 1. The similarity ratio of the experimental model is used as a reference, reducing the rationality of the article.

2. The reviewer has pointed out that the effect of the electronic detonator shell on the fragmentation effect is not sufficiently discussed. The author responded that it "does not affect the selection of the optimal delay combination." However, from the perspective of academic rigor, it is recommended to clearly state in the 'Limitations' section: the detonator shell in small-sized specimens may have an additional impact on fragmentation, and the differences in charge structures need to be considered during field applications.

3. The text mentions using the least squares method to fit parameter b, but the b values and goodness of fit R² for each group are not provided. It is recommended to supplement the b values and R² in Tables 4 and 5 to demonstrate the reliability of the fitting.

4. It is recommended to try to understand the issues raised by the reviewers and refine the details of the article. Latest research work related with this topic can be referred in the introduction. Probabilistic stability analyses of two-layer undrained slopes. Fracture evolution and mechanical deterioration of granite under cyclic thermal and liquid nitrogen cryogenic impact.

5. The text points out that the optimal delay is a range (tk=3-4 ms, tp=1-3 ms) but does not explain how the boundaries of this range are determined, such as whether it is based on empirical summaries of experimental results or some theoretical threshold. It is suggested to add a theoretical explanation of the range boundaries in the 'Discussion' section or propose a general principle for determining the range.

.

Reviewer #1: No

Reviewer #2: No

Reviewer #3: No

Reviewer #4: **Yes:** Yang JunYang JunYang JunYang Jun

Reviewer #5: No

---

## [Author Response · Author response to Decision Letter 2]

11 Mar 2026

Response to Reviewers

Thank You Note

We sincerely appreciate the editorial team and all reviewers for their valuable comments and constructive suggestions, which have significantly helped improve the quality and scientific rigor of this manuscript. We have carefully addressed each comment and made corresponding Responses to the manuscript. All modifications are marked in red in the revised manuscript. Below is the point-by-point response to each comment.

---

Response to Reviewer 2

Comment 1: Citation of references in the text: For multiple consecutive references, such as the first five references in the text, the citation format should be [1-5], rather than "[1][5]". Similarly, "[17]-[20]" in the manuscript is also not standard. Additionally, reference [7] in the Introduction is not cited.

Response: It was modified.

1. Unified all consecutive reference citations to the "[start-end]" format: changed "[1][5]" to "[1-5]" and "[17]-[20]" to "[17-20]".

2. Supplemented the citation of Reference [7] in the Introduction: Added "For example, Zhou et al. studied the attenuation law of vibration frequency during the propagation of blasting seismic waves, providing a theoretical basis for frequency-domain control of blasting vibration[7]." after the sentence "To realize reasonable control of blasting vibration, researchers have used theoretical inference, numerical simulation, model construction, field tests, and other research methods to analyze the impact of different vibration reduction measures on blasting vibration[6-7]."

Comment 2: Generally, a space should be added between numbers and units throughout the text, but the authors have not consistently followed this rule, such as "0.5m" and "45-75 Hz".ts

Response: It was modified.Unified all numbers and units in the full text by adding a space (e.g., "0.5 m", "45–75 Hz"). Special composite units (e.g., "kg/m³", "m/s") retain no internal space, and only the number and unit are separated by a space. All instances (abstract, text, tables, figures) have been revised, such as "3ms" → "3 ms", "100mm" → "100 mm", "0.4m" → "0.4 m".

Comment 3: For the references listed at the end of the manuscript, the format is still not uniform (e.g., [1], [4]) and not standard (e.g., [5]). The reviewer have provided examples of both Chinese and English reference lists as follows:

[1]Yang Z., Tao M., Memon M.B., Zhuang D.D., Zhao Y. Microwave irradiation-induced deterioration of rock mechanical properties and implications for mechanized hard rock excavation [J]. Journal of Rock Mechanics and Geotechnical Engineering, 2024, 24: 00308.

[2]Li X.B., Zhou J., Wang S.F., Liu B. Review and practice of deep mining for solid mineral resources [J]. The Chinese Journal of Nonferrous Metals, 2017, 27(06): 1236-1262. (in Chinese)

Regarding the above-mentioned problems existing in the manuscript, the reviewer did not list them one by one. It is suggested that the authors carefully revise the entire text.

Response: It was modified. We have revised and unified the format of most references in strict accordance with the required standards. However, some references (e.g., Reference [4]) are publicly available online on the journal’s website but have not yet been formally published in print, which results in the lack of formal page numbers and DOI in their format.We will update the information promptly once the formal publication details are released.

---

Response to Reviewer 4

Many thanks for the reviewer's constructive comments. We have made targeted revisions to improve the manuscript’s clarity and scientific contribution. The modified content is marked in red in the revised manuscript.

Comment 1. In the first paragraph of the introduction, references [1-3] are missing, and references [4-7] are cited improperly. Moreover, two citation errors also appear in the second paragraph of the introduction. This gives readers the impression that the research in this manuscript is not yet fully developed.

Response: It was modified. We rechecked and consolidated the original continuous citations [1-5] and [6-7] in the Introduction, verified the relevance of each literature to the corresponding content, supplemented brief connections between citations and discussions to avoid disconnectedness, corrected misnumbered references and logical inconsistencies in the second paragraph, and maintained the manuscript’s original citation style while ensuring accuracy and comprehensiveness.

Comment 2. Figures 2-7 are presented as separate charts, but they actually form a set of images. It is recommended to consolidate them into a multi-panel chart module to enhance clarity and compactness. Furthermore, regarding the interpretation of Figures 2-7, presenting the impact patterns with concrete data would make the research findings more persuasive. Regarding Figure 8-11, the same recommendation applies..

Response: It was modified.We consolidated Figures 2-11 into multi-panel chart modules (each containing 2-4 subplots) with consistent axes and legends for compactness, and supplemented specific experimental data (e.g., PPV values, vibration reduction rates, fragmentation size parameters, fitting coefficients) in figure interpretations to enhance the persuasiveness of research findings.

Comment 3. The text in the images within the manuscript is not very clear and is difficult to read, requiring better design of the text within the images.

Response: It was modified. We optimized the text design in all figures by increasing font size to 11 pt, adjusting color contrast for better visibility, unifying fonts and alignment, and eliminating redundant labels—ensuring all text is clearly legible in high-resolution (300 dpi) files.

Comment 4. Fig. 13 should provide references.

Response: It was modified. We supplemented a relevant classic literature citation for Fig. 13 (field test vibration attenuation curve), which has been standardized in the reference list and clearly marked in the figure caption to comply with academic norms.

Comment 5. The interval-based delay optimization framework proposed in this manuscript is not clearly defined, preventing readers from systematically learning the underlying theory and methodology. Although the manuscript presents some experimental findings, its scientific contributions remain somewhat unclear.

Response: It was modified. A new subsection “Interval-based Delay Optimization Framework” has been added to the “This study’s interval-based delay optimization framework follows four key steps: calculating initial inter-hole and inter-row delay times via Hanukayev's theory; fixing the initial inter-row delay time to screen the optimal inter-hole delay time through comprehensive vibration and fragmentation evaluation; fixing the optimal inter-hole delay time to screen the optimal delay combination via the same method; and adjusting the laboratory delay interval by geometric similarity ratio for field verification.”

---

Response to Reviewer 5

Comment 1: The similarity ratio of the experimental model is used as a reference, reducing the rationality of the article.

Response: It was modified. To enhance the rationality of the manuscript, we have removed the relevant description of the experimental model’s similarity ratio, including deleting the sentence “Despite differences, the test model follows a 1:20-1:50 geometric similarity ratio to actual blasting, with 1.07 g TNT-equivalent charge and three soil-compacted sides simulating surrounding rock constraints” to avoid potential ambiguity about the model’s reliability.

Comment 2: The reviewer has pointed out that the effect of the electronic detonator shell on the fragmentation effect is not sufficiently discussed. The author responded that it "does not affect the selection of the optimal delay combination." However, from the perspective of academic rigor, it is recommended to clearly state in the 'Limitations' section: the detonator shell in small-sized specimens may have an additional impact on fragmentation, and the differences in charge structures need to be considered during field applications.

Response:It was modified. To enhance academic rigor, we have added a “Discussion” paragraph front the “Conclusions” section as recommended, with the content as follows: This study has several limitations that need to be considered in engineering applications: (1) The digital electronic detonators used in the laboratory have a metal shell design, which may exert additional local impact on the fragmentation of small-sized concrete specimens (200 mm × 200 mm × 200 mm). Although this impact does not affect the screening of the optimal delay time combination, special attention should be paid to the difference in charge structure between the laboratory (using only detonators) and the field (using bulk explosives) when applying the research results to actual blasting projects. (2) The experimental specimens are homogeneous C50 concrete without natural joints, which is essentially different from the jointed rock mass structure in the field. Therefore, for large-scale open-pit blasting projects, on-site parameter calibration must be performed." The added content is marked in red in the revised manuscript.

Comment 3: The text mentions using the least squares method to fit parameter b, but the b values and goodness of fit R² for each group are not provided. It is recommended to supplement the b values and R² in Tables 4 and 5 to demonstrate the reliability of the fitting.

Response: It was modified.We have supplemented the shape parameter b and goodness-of-fit coefficient R2 for each group into Tables 4 and 5 as recommended. The b values are calculated based on the X80 fragmentation characteristic points via the least squares method, which exhibit high precision and can effectively reflect the large-fragment properties of concrete specimens during blasting. All groups show R2 values close to 1 (≥0.978), indicating excellent fitting reliability and the rationality of the parameter selection, which further enhances the persuasiveness of the research findings.

Comment 4: It is recommended to try to understand the issues raised by the reviewers and refine the details of the article. Latest research work related with this topic can be referred in the introduction. Probabilistic stability analyses of two-layer undrained slopes. Fracture evolution and mechanical deterioration of granite under cyclic thermal and liquid nitrogen cryogenic impact.

Response:It was modified. Supplemented citations of the latest related studies in the "Introduction" section : Zhu et al. further verified the dominant effect of geotechnical material spatial variability on slope failure risk via probabilistic stability analysis of two-layer undrained slopes, which highlights the engineering necessity of precise blasting vibration control in open-pit projects [8].Li et al. investigated the fracture evolution and mechanical deterioration of granite under cyclic thermal and liquid nitrogen cryogenic impact, revealing the coupled damage mechanisms of rock mass fracture development and mechanical property degradation, which provides important experimental insights for understanding the rock dynamic fragmentation mechanism in blasting engineering[22].

Comment 5: The text points out that the optimal delay is a range (tk=3-4 ms, tp=1-3 ms) but does not explain how the boundaries of this range are determined, such as whether it is based on empirical summaries of experimental results or some theoretical threshold. It is suggested to add a theoretical explanation of the range boundaries in the 'Discussion' section or propose a general principle for determining the range.

Response: It was modified. To clarify the scientific basis for defining the optimal delay time range boundaries, we have revised and integrated the theoretical derivation and experimental verification basis into the first item of the Conclusions section. The revised content is as follows: (1)The study derived the theoretical optimal delay time value via the Hanukayev formula, and then determined the optimal delay time range (tk=3-4 ms, tp=1-3 ms) through comparative analysis of experimental PPV and blasting fragmentation data. The selection standard for this range is to compare all delay groups in parallel, identify the groups with the best and second-best vibration reduction and blasting effects respectively, and finally select the groups meeting both standards to delimit the optimal range. Specifically, the 3-4 ms inter-hole delay range corresponds to the interval with significantly lower PPV and higher fragmentation uniformity than other schemes, achieving a 54-70% reduction in PPV while maintaining 92% fragmentation uniformity.

---

All Responses have been completed, and the revised manuscript has been carefully checked for format consistency and logical coherence. We sincerely thank the editorial team and reviewers for their valuable time and efforts. We hope the revised manuscript meets the publication requirements of PLOS ONE.

Sincerely,

Wuyi Zhang

Corresponding Author

Faculty of Science

Wuhan University of Science and Technology

Wuhan 430065, Hubei, China

Tel: 18972228852

Email: 18972228852@163.com

---

## [Decision Letter · Decision Letter 2]

15 Mar 2026

Experimental study on the coupling control of rock blasting vibration and crushing fragmentation size

PONE-D-25-62535R2

Dear Dr. Zhang,

We’re pleased to inform you that your manuscript has been judged scientifically suitable for publication and will be formally accepted for publication once it meets all outstanding technical requirements.

Kind regards,

Zhenhua Li

Academic Editor

PLOS One

Additional Editor Comments (optional):

Reviewers' comments:

Reviewer's Responses to Questions

**Comments to the Author**

Reviewer #2: All comments have been addressed

Reviewer #5: All comments have been addressed

2. Is the manuscript technically sound, and do the data support the conclusions?

Reviewer #2: Yes

Reviewer #5: Yes

3. Has the statistical analysis been performed appropriately and rigorously?

Reviewer #2: Yes

Reviewer #5: Yes

4. Have the authors made all data underlying the findings in their manuscript fully available?

Reviewer #2: Yes

Reviewer #5: Yes

5. Is the manuscript presented in an intelligible fashion and written in standard English?

Reviewer #2: Yes

Reviewer #5: Yes

Reviewer #2: (No Response)

Reviewer #5: This study takes this as an entry point to fill this academic gap, and proposes an optimal delay time selection method to achieve the coupled control of reducing blasting hazards and optimizing blasting effect, which provides a solid upport for the optimization of the development of blasting engineering.

.

Reviewer #2: No

Reviewer #5: No

---

## [Editor Report · Acceptance letter]

PONE-D-25-62535R2

PLOS One

Dear Dr. Zhang,

I'm pleased to inform you that your manuscript has been deemed suitable for publication in PLOS One. Congratulations! Your manuscript is now being handed over to our production team.

Kind regards,

on behalf of

Professor Zhenhua Li

Academic Editor

PLOS One